# You Only Train Once:
# Loss-Conditional Training of Deep Networks

**Alexey Dosovitskiy & Josip Djolonga**
Google Research, Brain Team
{adosovitskiy, josipd}@google.com

## Abstract

In many machine learning problems, loss functions are weighted sums of several terms. A typical approach to dealing with these is to train multiple separate models with different selections of weights and then either choose the best one according to some criterion or keep multiple models if it is desirable to maintain a diverse set of solutions. This is inefficient both at training and at inference time. We propose a method that allows replacing multiple models trained on one loss function each by a single model trained on a distribution of losses. At test time a model trained this way can be conditioned to generate outputs corresponding to any loss from the training distribution of losses. We demonstrate this approach on three tasks with parameterized losses: $\beta$-VAE, learned image compression, and fast style transfer.

## 1 Introduction

When designing a model for a machine learning problem at hand, a practitioner often thinks of multiple qualities that the model should have. For example, in addition to requiring a good prediction performance, it may be desirable to have a classifier that is simple, so that it would better generalize to unseen data. Similarly, when training a lossy image compression model, one optimizes both the size of the compressed images and their quality. In such scenarios, multiple loss functions have to be minimized simultaneously, each of them modelling a different facet of the considered problem.

Typically, it is not possible to simultaneously optimize all these losses — either due to limited model capacity, or because some of the losses may be fundamentally in conflict (for instance, image quality and the compression ratio in image compression). One thus has to decide how to balance the losses during optimization. Such multi-objective problems are most commonly scalarized by linearly combining the losses, with weights defining the trade-off between the loss terms. However, the exact values of the chosen weights can strongly affect the performance of the model and their tuning can be cumbersome. It might be also unpredictable how the weights affect the final values of the individual losses; moreover, the actual goal may be to optimize a down-stream metric that is not being explicitly minimized during training (for instance, classification accuracy). Furthermore, in many tasks there may not be a single optimal model, but it is instead important to make different trade-offs at test time. For instance, in image compression, to achieve a fine-grained coverage of rate-distortion trade-offs, one would first have to train at least ten models with different parameter settings, and then at inference time store them all in memory and select the appropriate one depending on the required compression level. This is inefficient both at training and at inference time. However, intuitively, the models trained for different loss variants are related and could share a large fraction of computation. Is it possible to improve the efficiency by making use of this model redundancy?

In this paper we propose a simple and broadly applicable approach that efficiently deals with multi-term loss functions and, more generally, arbitrarily parameterized loss functions. Instead of training one model for each parameter combination of the loss, we train one model that covers the whole space of different loss weightings. We achieve this by (i) training the model on a distribution of losses instead of a single loss, and (ii) conditioning the model outputs on the parameters of the

---

The code will be released at www.github.com/google-research/google-research/yoto.

loss function. This way, at inference time the conditioning vector can be varied, allowing us to traverse the space of models corresponding to different loss functions. We dub the method You Only Train Once (YOTO). The conceptual simplicity of the approach makes it easily applicable to many problem domains, with only minimal changes to existing codebases.

We experimentally showcase the efficacy of the proposed optimization scheme on image synthesis problems, where multi-loss problems are especially widespread. We first show that ($\beta$-)variational autoencoders (Kingma & Welling, 2014; Higgins et al., 2017) can be trained for a large range of $\beta$ parameters at the cost of training a single fixed-$\beta$ model. Then, we train a single deep image compression model (Ballé et al., 2018) that can adjust the rate-distortion trade-off on the fly at test time. Finally, we apply YOTO to style transfer (Gatys et al., 2016; Ghiasi et al., 2017), where it is provides a clear benefit given that there is no clear single optimum and that the effect of the various losses on the stylized image is difficult to characterize.

## 2 RELATED WORK

Losses with multiple terms are most commonly associated with multi-task learning (Caruana, 1997; Kokkinos, 2017; Zamir et al., 2018), where the goal is to train a single model that simultaneously solves several learning tasks. We refer the reader to Ruder (2017) for a recent review. Most related to our approach are methods that condition the network architecture on the task at hand (Rebuffi et al., 2017; Mallya et al., 2018), which an be seen as a special case of our method with "one hot" loss weight vectors. While ideas from our work could also be applied to the multi-task learning scenario, here we choose to focus on problems where it is desirable to learn a diverse set of models, rather than a single well-performing model. Recently, Brault et al. (2019) studied simultaneous optimization of multiple loss functions in the context of multi-task kernel learning. Our work is similar in spirit, but differs drastically both technically and in the application domains.

Conditional normalization schemes have recently been used for a range of supervised learning tasks, including Conditional Batch Normalization (CBN) for visual question answering (de Vries et al., 2017), Conditional Instance Normalization and Adaptive Instance Normalization (AdaIN) for fast style transfer (Dumoulin et al., 2017; Huang & Belongie, 2017; Ghiasi et al., 2017), Feature-wise Linear Modulation (FiLM) for visual reasoning (Perez et al., 2018), and a variety of conditioning schemes for generative models (Karras et al., 2019; Brock et al., 2019). A comprehensive overview of conditioning methods is provided by Dumoulin et al. (2018). All these methods condition a convolutional network on some side information by linearly transforming the activations with co-efficients computed from the side information. The side information typically informs the model about the prediction target: for instance, it can be the text of a question in the question answering setting or the style image in style transfer. The only exception we are aware of is the recent work of Babaeizadeh & Ghiasi (2018), which conditions a style transfer network on the parameters of the loss function, same as in our method. The approach is technically very similar, but we formulate it as a generally applicable technique and evaluate on several tasks both qualitatively and quantitatively.

Conditional models are also often used in sensorimotor control to train a single model capable of addressing a family of tasks, such as throwing darts at different targets (da Silva et al., 2012), block stacking and ball hitting (Deisenroth et al., 2014), navigating towards different locations (Schaul et al., 2015), moving objects with a robotic arm (Andrychowicz et al., 2017), driving in the desired direction (Codevilla et al., 2018) or trading off the reward components (Dosovitskiy & Koltun, 2017). A single model, conditioned on the task parameters, is trained to simultaneously maximize the rewards for a distribution of tasks. While technically similar to our setting, the conceptual difference is that between the tasks the training targets are changing (either rewards or expert actions used as labels for imitation learning), rather than the loss function itself, as in our case.

Another recent line of related work explores the use hypernetworks — networks that generate the weights of other networks — for hyperparameter optimization. Lorraine & Duvenaud (2018) propose to parametrize the weights of a network by a hypernetwork, which is conditioned on the values of the hyperparameters. The model can then be trained on a distribution of hyperparameters, and the hyperparameters can be optimized via gradient descent on the validation data. MacKay et al. (2019) improve the scalability of the method by proposing efficient approximations to hypernet-works. These methods are technically similar to our approach, but they focus on hyperparameter

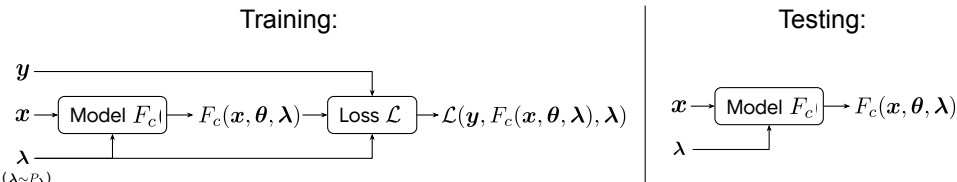

Figure 1: Illustration of the proposed method. At training time (left), for each training data point loss parameters $\boldsymbol{\lambda}$ are sampled from a distribution $P_{\boldsymbol{\lambda}}$, and the model is conditioned on these. Then, at test time (right), the model can be conditioned on any desired loss parameters to achieve behavior corresponding to the loss function with these parameters.

optimization, while we are more interested in obtaining the full set of solutions corresponding to the different loss coefficients.

Recently, Shoshan et al. (2019) and Wang et al. (2019) addressed the problem of interpolating between predictions of two or more image generation models. Both methods start by training several separate models and then propose different ways of interpolating between them: either by using specialized adaptor blocks in the networks, or by directly interpolating the network weights. We, in contrast, train only a single model by directly optimizing the performance over a distribution of losses. This is conceptually simpler, can be easily integrated in existing models, and requires training a single model instead of several.

## 3 METHOD

We consider the following learning scenario: given a training distribution of pairs $\boldsymbol{x}, \boldsymbol{y} \sim P_{\boldsymbol{x},\boldsymbol{y}}$ with $\boldsymbol{x} \in \mathbb{X} \subset \mathbb{R}^{d_{\mathbb{X}}}$, $\boldsymbol{y} \in \mathbb{Y} \subset \mathbb{R}^{d_{\mathbb{Y}}}$, and a loss function $\mathcal{L}(\cdot, \cdot) : \mathbb{Y} \times \mathbb{Y} \to \mathbb{R}$, we aim to learn a model $F : \mathbb{X} \to \mathbb{Y}$, with parameters $\boldsymbol{\theta}$, such that its predictions $\hat{\boldsymbol{y}} = F(\boldsymbol{x}, \boldsymbol{\theta})$ minimize the expected value of the loss $\mathcal{L}(\boldsymbol{y}, \hat{\boldsymbol{y}})$ over the dataset. This formulation is very broad and covers a variety of machine learning problems, including (self-)supervised learning learning, and some types of generative models. We focus on this setting for a more streamlined presentation, although the proposed method can be applied to models that go beyond this formulation.

Instead of a single fixed loss function $\mathcal{L}(\cdot, \cdot)$, assume that we are interested in a family of losses $\mathcal{L}(\cdot, \cdot, \boldsymbol{\lambda})$, parameterized by a vector $\boldsymbol{\lambda} \in \Lambda \subset \mathbb{R}^{d_{\boldsymbol{\lambda}}}$. The most common case of such a family of losses is a weighted sum of several loss terms: $\mathcal{L}(\cdot, \cdot, \boldsymbol{\lambda}) = \sum_i \lambda_i \mathcal{L}^i(\cdot, \cdot)$. However, other parameters, such as, for instance, the power $p$ in $\ell^p$ losses, might also be included in the vector $\boldsymbol{\lambda}$. In practice, one then typically minimizes the loss independently for each choice of the parameter vector $\boldsymbol{\lambda}$:

$$\boldsymbol{\theta}^*_{\boldsymbol{\lambda}} = \arg\min_{\boldsymbol{\theta}} \mathbb{E}_{\boldsymbol{x},\boldsymbol{y}\sim P_{\boldsymbol{x},\boldsymbol{y}}} \mathcal{L}(\boldsymbol{y}, F(\boldsymbol{x}, \boldsymbol{\theta}), \boldsymbol{\lambda}). \tag{1}$$

Instead of fixing $\boldsymbol{\lambda}$, we propose to solve an optimization problem where the parameters $\boldsymbol{\lambda}$ are sampled from a distribution $P_{\boldsymbol{\lambda}}$. Hence, during training time the model observes many losses, and can learn to utilize the relationships between them to optimize all of them simultaneously. The distribution $P_{\boldsymbol{\lambda}}$ can be seen as a prior over the losses, and we discuss its choice in the next subsection. At inference time, the joint model can be conditioned on an arbitrary desired parameter value $\boldsymbol{\lambda}$, yielding the corresponding predictions. We thus solve the following optimization problem:

$$\boldsymbol{\theta}^* = \arg\min_{\boldsymbol{\theta}} \mathbb{E}_{\boldsymbol{\lambda}\sim P_{\boldsymbol{\lambda}}} \mathbb{E}_{\boldsymbol{x},\boldsymbol{y}\sim P_{\boldsymbol{x},\boldsymbol{y}}} \mathcal{L}(\boldsymbol{y}, F_c(\boldsymbol{x}, \boldsymbol{\theta}, \boldsymbol{\lambda}), \boldsymbol{\lambda}). \tag{2}$$

In the limit of infinite model capacity, this new proposed optimization problem is equivalent to the original independent optimization scheme, in the sense that the minimal values for both problems coincide. Further discussion, as well as the proof, are provided in the appendix.

**Proposition 1.** *Consider two optimization problems:*

$$(1) \min_{F\in C(\mathbb{X})} \mathbb{E}_{\boldsymbol{x},\boldsymbol{y}\sim P_{\boldsymbol{x},\boldsymbol{y}}} \mathcal{L}(\boldsymbol{y}, F(\boldsymbol{x}), \boldsymbol{\lambda}) \qquad (2) \min_{G\in C(\mathbb{X}\times\Lambda)} \mathbb{E}_{\boldsymbol{\lambda}\sim P_{\boldsymbol{\lambda}}} \mathbb{E}_{\boldsymbol{x},\boldsymbol{y}\sim P_{\boldsymbol{x},\boldsymbol{y}}} \mathcal{L}(\boldsymbol{y}, G(\boldsymbol{x}, \boldsymbol{\lambda}), \boldsymbol{\lambda}),$$

*where $C(S)$ denotes the space of continuous functions on $S$. Assume there exists a continuous function $F^*(\boldsymbol{x}, \boldsymbol{\lambda})$ such that for every $\boldsymbol{\lambda} \in \Lambda$ the function $F^*(\cdot, \boldsymbol{\lambda})$ solves the problem (1). Then, if $G^*$ is a solution of the problem (2), $G^*(\cdot, \lambda)$ also minimizes the problem (1) almost surely w.r.t. $P_{\boldsymbol{\lambda}}$.*

This theoretical result shows that the method is fundamentally as powerful as the standard per-parameter training, while only requiring a single model. However, in practice the infinite capacity assumption does not hold, so implementation details become important, which we describe next.

## 3.1 PRACTICAL DETAILS

In this work we focus on training deep networks with methods based on stochastic gradient descent. We therefore approximate the expectations in equation 2 using Monte Carlo estimates: we estimate the expectation over the data distribution by averaging over a training set $D = \{(\boldsymbol{x}_i, \boldsymbol{y}_i)\}_{i=1}^N$, and the expectation over the loss parameters by sampling from the loss parameter distribution:

$$\boldsymbol{\theta}^* = \arg\min_{\boldsymbol{\theta}} \sum_{i=1}^n L(\boldsymbol{y}_i, F(\boldsymbol{x}_i, \boldsymbol{\theta}, \boldsymbol{\lambda}_i), \boldsymbol{\lambda}_i), \qquad \boldsymbol{\lambda}_i \sim P_{\boldsymbol{\lambda}}. \tag{3}$$

In our experiments, we sample a new loss parameter for each training data point every time it is encountered during training.

The distribution of the loss parameters to be used for training is typically not given and has to be chosen. Clearly, the support of the training distribution has to include the range of parameter values we are interested in covering, but this still leaves a lot of freedom in the specific choice of the distribution. In our experiments we use the log-uniform distribution, which is commonly used for sampling the parameters of machine learning models (Bergstra & Bengio, 2012).

In our experiments the model $F$ is a convolutional network. To condition the network on the loss parameters, we use Feature-wise Linear Modulation (FiLM) (Perez et al., 2018). First, we select the layers of the network to be conditioned (can be all layers or a subset). Next, we condition each of the layers on the given weight parameters $\boldsymbol{\lambda}$. Assume the layer outputs a feature map $\boldsymbol{f}$ of dimensions $W \times H \times C$, with $W$ and $H$ corresponding to the spatial dimensions and $C$ to the channels. We feed the parameter vector $\boldsymbol{\lambda}$ to two multi-layer perceptrons (MLPs) $M_\sigma$ and $M_\mu$ to generate two vectors, $\boldsymbol{\sigma}$ and $\boldsymbol{\mu}$, of dimensionality $C$ each. We then multiply the feature map channel-wise by $\boldsymbol{\sigma}$ and add $\boldsymbol{\mu}$ to get the transformed feature map $\tilde{\boldsymbol{f}}$:

$$\tilde{f}_{ijk} = \sigma_k f_{ijk} + \mu_k, \quad \boldsymbol{\sigma} = M_\sigma(\boldsymbol{\lambda}), \ \boldsymbol{\mu} = M_\mu(\boldsymbol{\lambda}). \tag{4}$$

## 4 EXPERIMENTS

We evaluate the proposed method both quantitatively and qualitatively on three problems with multi-term loss functions: $\beta$-VAE, learned image compression, and fast style transfer. All details regarding the architectures and the training, as well as additional results, are provided in the appendix.

## 4.1 $\beta$-VARIATIONAL AUTOENCODERS

Our first problem is the loss of the $\beta$-variational autoencoder (Higgins et al., 2017)

$$\mathcal{L}_{\beta\text{-VAE}}(\boldsymbol{x}, \boldsymbol{z}, \phi, \theta, \beta) = \mathbb{E}_{q_\phi(\boldsymbol{z}|\boldsymbol{x})} \log p_\theta(\boldsymbol{x} \mid \boldsymbol{z}) - \beta D_{\text{KL}}(q_\phi(\boldsymbol{z} \mid \boldsymbol{x}) \| p(\boldsymbol{z})), \tag{5}$$

where $q_\phi(\boldsymbol{z} \mid \boldsymbol{x})$ is the amortized approximate posterior implemented by an encoder network with parameters $\phi$, $p_\theta(\boldsymbol{x} \mid \boldsymbol{z})$ is a stochastic decoder network with parameters $\theta$, and $p(\boldsymbol{z})$ is the prior distribution, in our case a standard multivariate Gaussian. The loss has a single positive parameter $\beta$ trading off between the reconstruction quality and the divergence between the approximate posterior and the prior.

We consider two settings: the CIFAR-10 dataset (Krizhevsky, 2009) with Gaussian outputs, and the Shapes3D dataset (Burgess & Kim, 2018) with Bernoulli outputs. In both cases we use convolutional encoder-decoder networks with Gaussian latents. We condition after each downsampling or upsampling operation. Both for YOTO and fixed-weight models, we train a set of networks with varying capacities, by proportionally changing the width of all layers. This is done with the goal

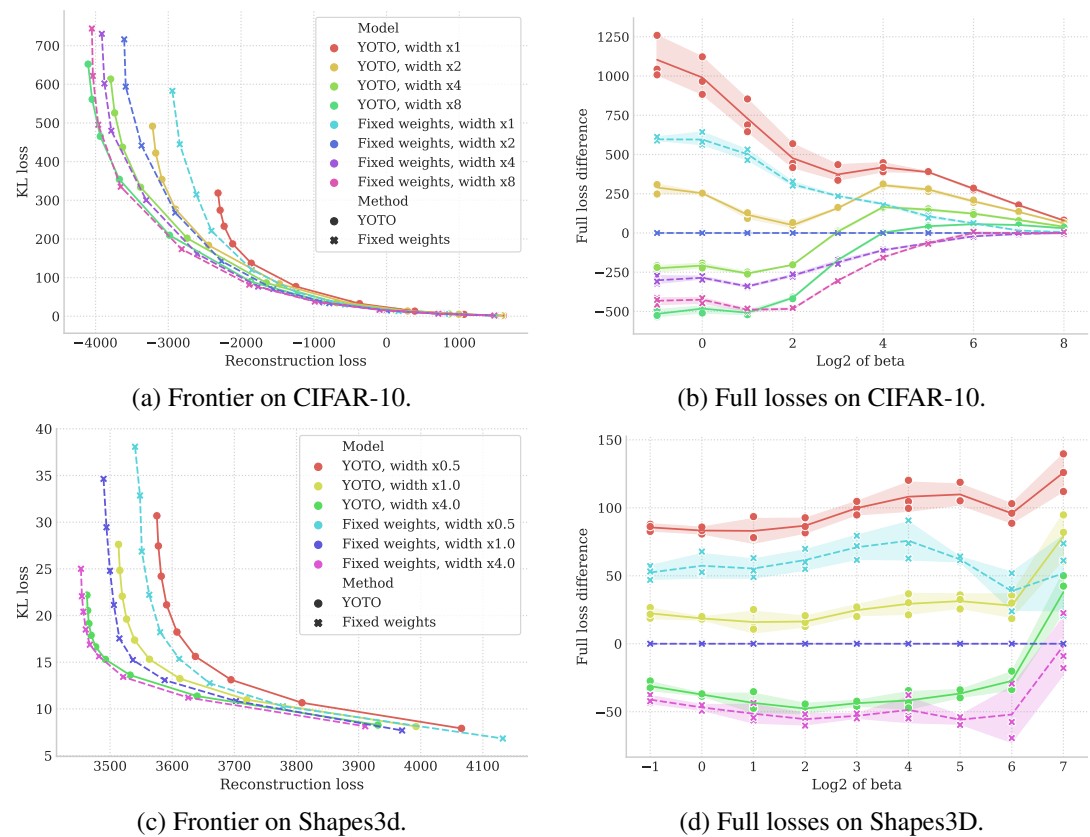

Figure 2: Quantitative $\beta$-VAE results on CIFAR-10 (a,b), and Shapes3D (c,d) for models of varying capacity (width). In most cases, the proposed method performs close to models trained independently for each loss weight value. This is especially the case for high capacity models.

of measuring how does the capacity of a YOTO model affect its performance relative to a set of fixed-weight models. We train each network 3 times and report means and standard deviations over these three runs.

Quantitative results are shown in Figure 2. For each dataset, we provide two plots: the frontier of the two loss components plotted against each other ((a), (c)), and the value of the full loss plotted against the value of the parameter $\beta$ ((b), (d)). The first plot allows to assess how well the proposed model fits the two loss terms, while the second one shows the actual optimization objective — the weighted sum of the losses. To better highlight the differences between the methods, when plotting the full loss, we normalize it by subtracting the loss of one of the models from all other models.

On both datasets, we observe several common tendencies. First, for small-capacity models the proposed method somewhat underperforms relative to the fixed-weight models, especially on CIFAR-10. This is not surprising, since it is difficult for a small model to cover all loss variants. Second, both the YOTO models and the fixed-weight ones improve substantially with increasing model capacity, so a set of fixed-weight models of a certain width is consistently matched or outperformed by a twice wider YOTO model (this is additionally highlighted in Figure 4 that plots the average full loss versus the width of the network). Third, the larger the network capacity, the closer the performance of fixed-models and the YOTO model. In particular, on both datasets for the widest networks the two frontiers become very close. This is quite intuitive and matches the theoretical result that in the limit of infinite model capacity the proposed method should be as powerful as training separate fixed-weight models.

Qualitative results are shown in Figure 3. Both for reconstruction and sampling, for each value of the weight $\beta$ that we show YOTO produces results closely resembling those of the corresponding

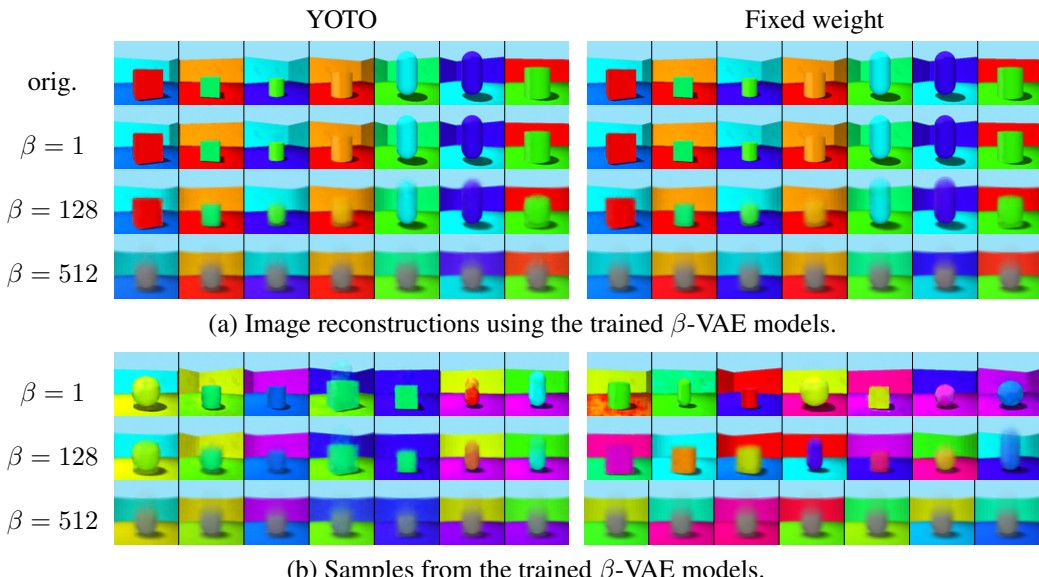

(a) Image reconstructions using the trained $\beta$-VAE models.

(b) Samples from the trained $\beta$-VAE models.

Figure 3: Qualitative $\beta$-VAE results on the Shapes3D dataset. For each loss weight $\beta$, the reconstruction and sampling results of YOTO are very similar to those of the separately trained models.

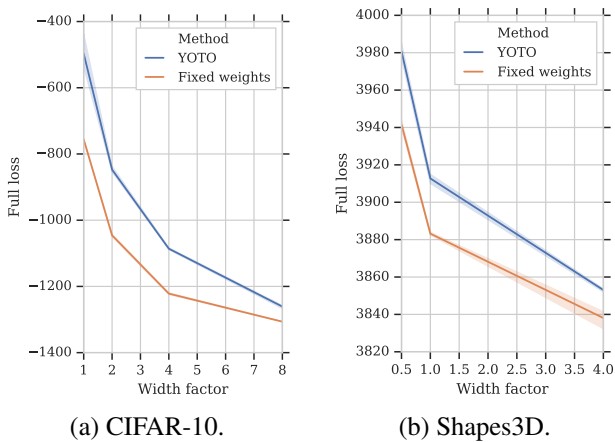

(a) CIFAR-10.  (b) Shapes3D.

Figure 4: Loss averaged over all loss weight values as a function of the model capacity. We compare the proposed method to a set of fixed-weight models on two datasets: CIFAR-10 (a) and Shapes3D (b). We vary the capacity of the network by multiplying the width of the network by a factor. The performance of the proposed method overall matches that of fixed-based models with a roughly 1.5x wider model. Moreover, the performance of the proposed method gets closer to that of the fixed-weight models when the capacity of the model is increased.

independently trained model. An interesting property of the YOTO model is that sampling with the same noise vector for different beta values generates similar images at different quality levels.

Additional results, including a comparison to two simple baselines (a single fixed-weight model and a baseline based on interpolation), an ablation study of the proposed method, and qualitative results on CIFAR-10, are shown in the appendix.

## 4.2 LEARNED IMAGE COMPRESSION

We build on the learned compression model of Ballé et al. (2018). The model is similar to a variational autoencoder, but with a few modifications that allow for the quantization of the latent representation and its use as a compressed image encoding. Moreover, it includes a "hyperprior" learned

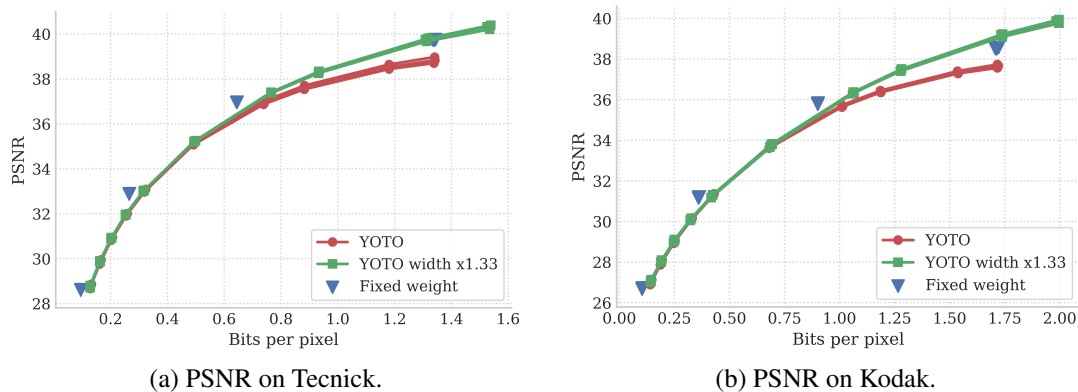

(a) PSNR on Tecnick.  (b) PSNR on Kodak.

Figure 5: Quantitative compression results on the Tecnick and Kodak datasets. A basic YOTO model under-performs compared to a set of fixed-weight models, but a wider network nearly matches the performance of the fixed-weight models, especially in the high-quality regime.

by a second variational autoencoder on top of the latents of the first one. The loss function has a single parameter — the "rate-distortion weight", trading off compression rate versus reconstruction quality. We refer the reader to (Ballé et al., 2018) for details. The architecture of the model includes four networks: a convolutional encoder and hyper-encoder, as well as a transposed convolutional decoder and a hyper-decoder. The networks are relatively shallow, with three convolutional layers each. We apply conditioning to all layers of the model. Same as for VAEs, we train three models for each setting to estimate the stability of training. Further details are provided in the appendix.

We evaluate the compression models on two datasets: Kodak (Kodak, 1993) and Tecnick (Asuni & Giachetti, 2014). The quantitative results on Tecnick are shown in Figure 5, while the results on Kodak are deferred to the appendix. We show rate-distortion plots with standard PSNR on pixel values as a measure of compression quality and bits per pixel as a measure of compression rate. We plot three models per method, but since the training is stable, there difference between them is almost not visible. The gap between separate fixed-weight models ("Fixed weight") and a single joint model of the same size ("YOTO") is fairly small, but consistent across all compression rates and is especially pronounced in the higher quality regime. However, making the model larger ("YOTO width x1.33") recovers a large fraction of the performance, in particular the wider model matches the performance of a smaller fixed-weight model in the high quality regime. However, in the high-compression regime there is still a small gap from the fixed-weight models.

We hypothesize that the reason for relatively worse performance of YOTO on the compression task compared to $\beta$-VAEs is that the networks used by Ballé et al. (2018) are relatively shallow. They therefore lack the representational power to efficiently make use of the conditioning input. We expect that one could devise combinations of architectures and conditioning methods for the compression task which would lead to a smaller gap between the fixed-weight and YOTO models; however, this additional tuning is beyond the scope of this paper.

### 4.3    STYLE TRANSFER

The final problem we consider is style transfer (Gatys et al., 2016) — given a content image $x_c$ and a style image $x_s$, the goal is to synthesize a stylized image $y$ that contains the same content as $x_c$, but in a style that resembles that of $x_s$. This is typically cast as an optimization problem — we want to find an image $y$, that when passed through a pre-trained neural network has activations similar to those of $x_c$, but whose aggregate feature statistics resemble those of $x_s$. There is a lot of freedom in deciding which layers of the network to use and how to weigh them. These parameters are not easy to tune, since we are optimizing for visual appearance, which cannot be easily quantified. Hence, a single model capable of representing all loss variants would be of great benefit. These experiments are similar to those presented by Babaeizadeh & Ghiasi (2018), but with more focus on a quantitative evaluation, same as in other applications reported above.

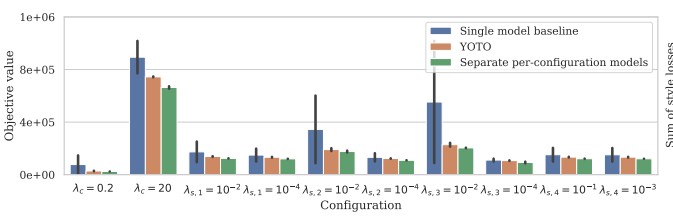 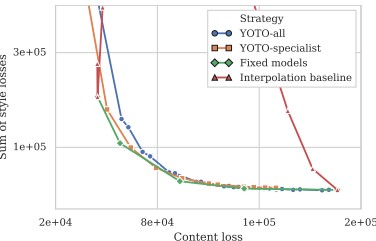

(a) Performance of YOTO when we unilaterally vary each weight. We show means and standard deviations over four runs.

(b) Sum of all style losses plotted against the content loss.

Figure 6: Quantitative results on style transfer. In both cases YOTO performance approaches or matches that of separate models trained per loss parameter vector.

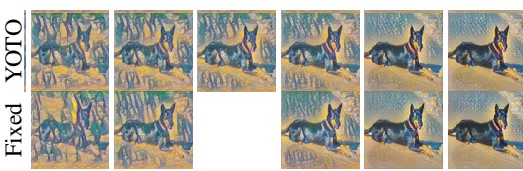 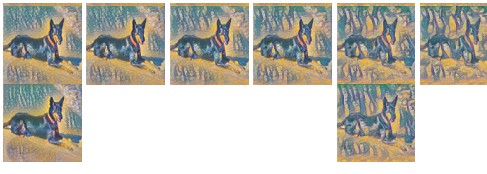

(a) Varying the content coefficient $\lambda_c$

(b) Varying one of the style coefficients $\lambda_{s,3}$

Figure 7: Qualitative comparison of image stylization models on an image from the validation set of ImageNet. The first row shows the results of YOTO trained on all parameters, and the second row shows the results of fixed-weight models trained independently for each loss parameters. In both cases the parameter values increase from left to right. YOTO generates results very similar to the separate models, while training just a single model.

We build on the work of Ghiasi et al. (2017), who propose training a single deep network that can stylize images in arbitrary styles. To this end, the content image is fed to the stylization network itself, while the style image is fed to a different "style prediction" network that extracts a feature embedding. This embedding is then used to modulate the stylization network via conditional instance normalization. We refer the reader to (Ghiasi et al., 2017) for a detailed explanation.

Following Ghiasi et al. (2017), we extract the activations from the VGG network (Simonyan & Zisserman, 2015), and define a total of six losses: four style losses $\mathcal{L}_{s,1}(\boldsymbol{y}, \boldsymbol{x}_s), \ldots, \mathcal{L}_{s,4}(\boldsymbol{y}, \boldsymbol{x}_s)$, one content loss $\mathcal{L}_c(\boldsymbol{y}, \boldsymbol{x}_c)$, and a total variation loss $\mathcal{L}_{\text{tv}}(\boldsymbol{y})$ encouraging smoothness between the neighbouring pixels of $\boldsymbol{y}$. The final loss is then a linear weighting of these, which we will denote as $\mathcal{L}(\boldsymbol{y}) = \sum_{i=1}^{n} \lambda_{s,i} \mathcal{L}_{s,i}(\boldsymbol{y}, \boldsymbol{x}_s) + \lambda_c \mathcal{L}_c(\boldsymbol{x}, \boldsymbol{c}) + \lambda_{\text{tv}} \mathcal{L}_{\text{tv}}(\boldsymbol{y})$. We train a single YOTO model over five of these parameters (while keeping the TV loss fixed to $10^4$), with log-uniform distributions. To this end, in addition to the style embeddings we condition the stylization network on the vector of loss parameters $\boldsymbol{\lambda}$. We sample the content images form ImageNet (Deng et al., 2009) and use 14 pointillism paintings as the style images. Further details are provided in the appendix.

We start by reporting quantitative results. As the problem has 5 loss parameters, it is not possible to visualize the frontier with respect to all losses. Instead, we take two visualization approaches. First, we show the full loss achieved by different methods on a set of loss parameter values selected as follows: we start with the default parameter settings of Ghiasi et al. (2017) $\lambda_{s,1} = \lambda_{s,2} = \lambda_{s,3} = 10^{-3}, \lambda_{s,4} = 10^{-2}, \lambda_c = 2$, and then either increase or decrease one of these parameter by a factor of 10 while keeping the rest fixed. This adds up to a total of 10 parameter values: 2 variants per each of the 5 parameters. To better understand the variability of the learned models due to the noisy training process we further trained each model four times. We provide the results in Figure 6a. As a baseline, for each parameter configuration we report the full loss value for the model trained with the default parameters ("Single model baseline"). The YOTO model trained on all parameters closely matches the performance of fixed-weight models trained for each of the parameter configurations.

Next, we focus on just one aspect of the loss surface: the trade-off between the content loss and the sum of style losses. In Figure 6b we plot the frontier of these two values corresponding to different

methods. Here we add a YOTO "specialist" model to the comparison, which has been trained with only 1 parameter, the content loss $\lambda_c$. Moreover, we report a baseline method that simply interpolates the images produced by two models trained with the extreme values of the content weight. Both YOTO models are close to the fixed-wight models, with the frontier of the "specialist" model closely tracing the fixed models. The interpolation baseline performs very poorly.

Finally, we show select qualitative results in Figure 7, where we compare fixed weight models to a YOTO model trained on all five parameters. We vary the content weight $\lambda_c$ and the third style weight $\lambda_{s,3}$, which contribute most to the full loss. We can see that YOTO captures the effect of changing each of these weights and obtains results similar to those of the separately trained models.

## 5 CONCLUSION

We have presented an approach to training a single deep model to fit a parametric family of loss functions. This way, instead of training multiple models corresponding to different loss variants, we can train a single model that provides solutions for all loss variants. We demonstrated the successful application of the method on three applications: $\beta$-VAE, learned image compression, and fast style transfer. These initial results are promising, but several challenges remain. First, it would be interesting to devise provably effective strategies of selecting the loss parameter distribution, especially for losses with multiple parameters. Second, further experiments with various architectures and conditioning methods can lead to an even smaller discrepancy between the separate models and our YOTO model. Third, the method could be used for many other applications, within or beyond the domain of image generation. Fourth, one might use YOTO-trained models to initialize usual fixed-loss training or, vice-versa, use a pre-trained fixed-loss model to initialize the training of YOTO. Finally, we believe that this work can also open a new family of theoretical questions centered around the analysis of problems that have to optimize a continuum of loss functions. We see all these directions as exciting avenues for future research.

## ACKNOWLEDGEMENTS

We sincerely thank Johannes Ballé for the help with setting up the image compression experiments and insightful discussions. We also thank Sebastian Nowozin, Rodolphe Jenatton, Joan Puigcerver, Sylvain Gelly, and others in Google Brain for useful discussions and the support of the project.

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

APPENDIX

## A ARCHITECTURE AND TRAINING DETAILS

### A.1 BETA-VAE

We use convolutional encoder-decoder networks on both datasets. We use leaky rectified linear layer (ReLU) non-linearity in all networks, except for units predicting stangard deviation of Gaussians where we use softmax non-linearity. On CIFAR-10 the network inputs have size $32 \times 32$ pixels, on Shapes3D – $64 \times 64$ pixels. We now describe the base network architectures, corresponding to "width x1.0".

On CIFAR-10 both the encoder and the decoder consist of 3 blocks of 2 convolutional layers each, with the first layer in each block having stride 2. All kernels have spatial size $3 \times 3$, and the number of channels is, respectively, 8, 8, 16, 16, 32, 32 (and increased proportionally for the "wider" models). The convolutional encoder is followed by fully connected layers with 256 and 512 units respectively, where the output of the latter layer is split in two 256-dimensional vectors encoding the mean and the variance of the approximate posterior. The decoder starts two fully connected layers with 256 and 512 units, the output of the latter layer is reshaped to a $4 \times 4 \times 32$ feature map and further processed by a decoder that is symmetric to the encoder.

On Shapes3D both the encoder and the decoder consist of 4 convolutional layers each, all with stride 2. All kernels have spatial size $3 \times 3$, and the number of channels is, respectively, 8, 16, 32, 64 (and increased proportionally for the "wider" models). The convolutional encoder is followed by fully connected layers with 256 and 20 units respectively, where the output of the latter layer is split in two 10-dimensional vectors encoding the mean and the variance of the approximate posterior. The decoder starts two fully connected layers with 256 and 1024 units, the output of the latter layer is reshaped to a $4 \times 4 \times 64$ feature map and further processed by a decoder that is symmetric to the encoder.

On Shapes3D we train all models for a total of $600,000$ mini-batch iterations, and we multiply the learning rate by 0.5 after $300,000$, $390,000$, $480,000$, and $570,000$ iterations. On CIFAR-10 we use a proportionally twice longer schedule. We tuned the learning rates by sweeping over the values $\{5 \cdot 10^{-5}, 1 \cdot 10^{-4}, 2 \cdot 10^{-4}, 4 \cdot 10{-4}, 8 \cdot 10^{-4}\}$ and ended up using the learning rates $1 \cdot 10^{-4}$ on CIFAR-10 and $2 \cdot 10^{-4}$ on Shapes3D. We use mini-batches of 128 samples on CIFAR-10 and 64 samples on Shapes3D. We use weight decay of $10^{-5}$ in all models.

For YOTO models, we condition the last layer of each convolutional block. The conditioning MLP has one hidden layer with 256 units on Shapes3D and 512 units on CIFAR-10. At training time we sample the $\beta$ parameter from log-normal distribution on the interval $[0.125, 1024.]$ for Shapes3D and on the interval $[0.125, 512.]$ for CIFAR-10.

### A.2 IMAGE COMPRESSION

We train the model as proposed by Ballé et al. (2018), and we refer the reader to that paper for architecture details. We train the models with the standard MSE reconstruction loss. All models have 192 channels in all layers, except for the "width x1.33" ones that have 256 channels. We use mini-batches of 8 samples. We train for 2 million mini-batch iterations with the learning rate of $10^{-4}$. We condition all convolutional layers in all networks using MLPs with 1 hidden layer of 128 units. During training we sample the rate-distortion weight log-uniformly from the interval $[1.2 \cdot 10^{-3}, 2.6 \cdot 10^{-1}]$.

### A.3 FAST STYLE TRANSFER

We build on the work of Ghiasi et al. (2017), and use the exact same network architecture and training protocol. We optimize for 2 million steps with a learning rate of $10^{-5}$. The ranges that we used were $[0.1, 100]$ for the content parameters, $[10^{-4}, 10^{-1}]$ for the first three style parameters and $[10^{-3}, 1]$ for the fourth one — we always sampled from a log-uniform distribution.

We condition the models by 1) sending the logarithms of the weights to an MLP with a single hidden layer of size 512 with ReLu activations and an output dimension of size, 2) concatenating these inputs to the inception features of the style image, 3) for each layer modulation we use an MLP

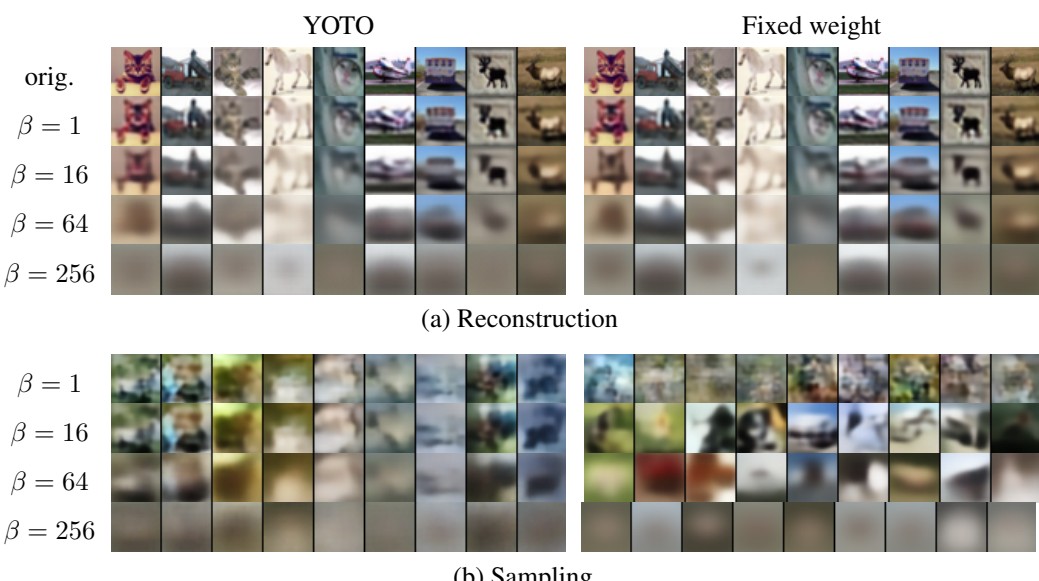

(a) Reconstruction

(b) Sampling

Figure 8: Qualitative VAE results on the CIFAR-10 dataset. Both for reconstruction and sampling, the YOTO results are qualitatively very similar to those of separate models trained for each value of the loss weight.

with a single layer of dimension 256 and ReLU activations, on which two affine maps are applied to compute the scale and shift.

## B    ADDITIONAL RESULTS

### B.1    BETA-VAE

Figure 8 shows qualitative results for $\beta$-VAE on CIFAR-10. Same as for Shapes3D, both reconstructions and samples from the YOTO model are qualitatively very similar to those generated with models trained separately for each weight.

We compare the proposed method to two baselines. The first one is a naive baseline — training a VAE with a fixed $\beta$, and computing the loss value of this fixed model for all other $\beta$ values ("Single model baseline"). We select the fixed $\beta$ so that it minimizes the average validation loss over all $\beta$ values. The second baseline interpolates between two models trained with fixed weights. Namely, we train two beta-VAEs for the extreme values of beta, and interpolate both the latents and the reconstructions produced by these models. We then measure the KL and reconstruction losses for these interpolated values. Note that this interpolation baseline is not a VAE any more: the latents and the reconstructions are averaged independently, so the interpolated reconstruction cannot be generated from the interpolated latent. Therefore, it is not clear how would one sample from such a model, and we do not report these results in the main paper to avoid confusion.

The results are shown in Figure 9. Surprisingly, on Shapes3D the baseline performs very close to the fixed-model and YOTO frontiers, and sometimes even outperforms them. In contrast, on CIFAR-10 the baseline is much worse than the other methods. We believe the reason is that while averaging the logits for Bernoulli outputs on Shapes3D is quite meaningful, averaging the standard deviations of Gaussian outputs on CIFAR-10 strongly affects Gaussian log-likelihood.

**Timing.**    We timed the training of a fixed-weight $\beta$-VAE and a YOTO model. To this end, we trained both models on a single CPU core over 200 mini-batch iterations. We found the YOTO model to be only 8% slower than its fixed-weight counterpart: 4.64 vs 5.04 training steps per second with the "width x2" network architecture.

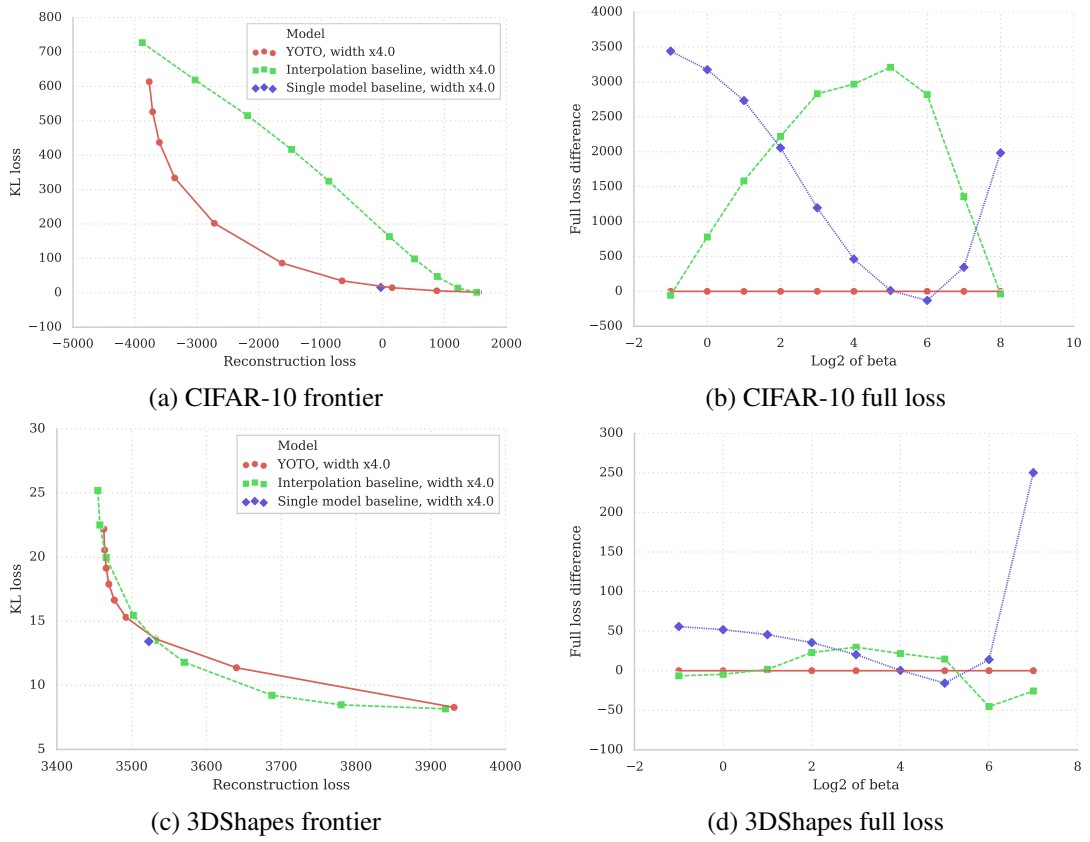

(a) CIFAR-10 frontier

(b) CIFAR-10 full loss

(c) 3DShapes frontier

(d) 3DShapes full loss

Figure 9: Quantitative comparison of the proposed method with baselines for beta-VAE on CIFAR-10 (a,b), and Shapes3D (c,d). The proposed method outperforms the "single model" baseline by a large model. The "interpolation" baseline is reported for completeness, but it is not a fair baseline, since it does not correspond to a VAE.

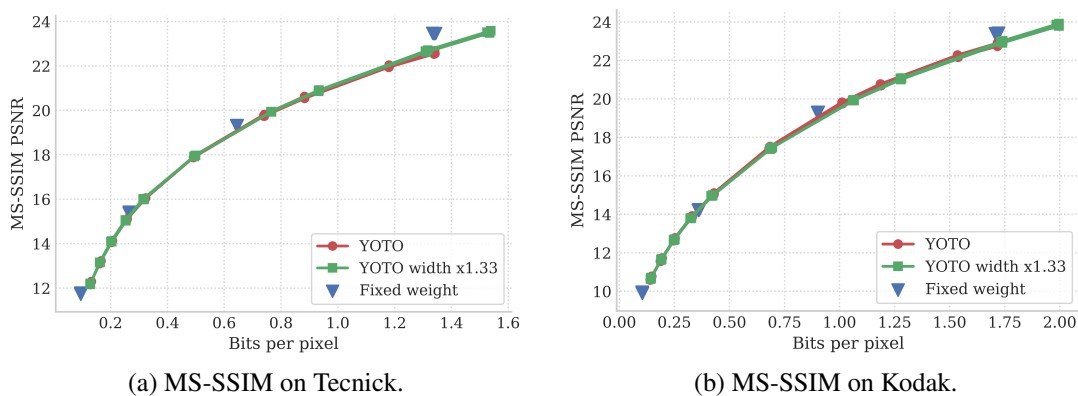

(a) MS-SSIM on Tecnick.

(b) MS-SSIM on Kodak.

Figure 10: Quantitative results of compression with the MS-SSIM metric on Tecnick (a) and Kodak (b).

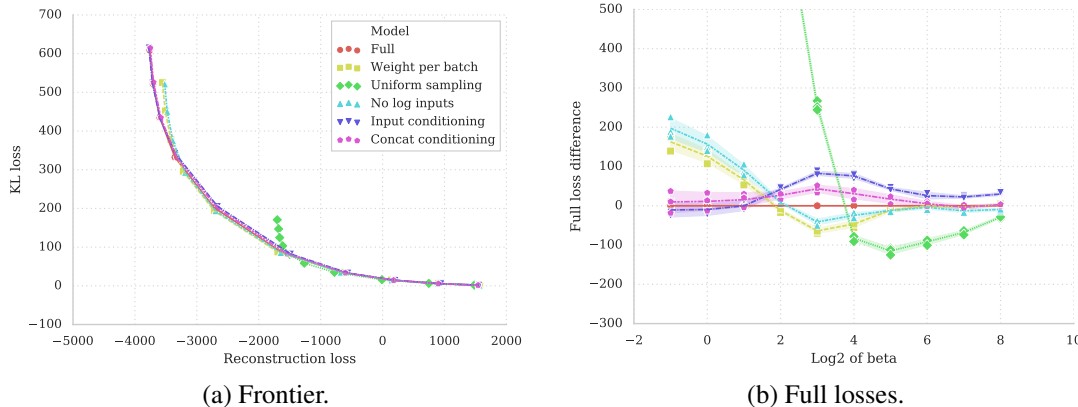

(a) Frontier.          (b) Full losses.

Figure 11: Ablation study of the CIFAR-10 $\beta$-VAE. The distribution of the loss weight used at training time has the largest impact on the performance. Moreover, conditioning the model by providing the loss weights as inputs to the model underperforms compared to more advanced conditioning methods. Other variants all perform close to the full method.

## B.2   ABLATION STUDY

We now ablate different design choices made in our implementation of YOTO to check their importance. We perform this analysis on the CIFAR-10 dataset, with the "width x4" architecture. The design decisions we study are: the weight distribution used during training (log-uniform or uniform), the method of sampling the loss weights (per sample or per mini-batch), the method of conditioning the network on the weights (FiLM, concatenating the conditioning vectors with the feature maps, or feeding the conditioning vectors to the network as an additional input), pre-processing of the conditioning inputs (taking their logarithm or not).

The results are presented in Figure 11. Clearly, the choice of the parameter distribution during training is the most important choice, and log-uniform distribution performs much better than uniform. Other choices are of less importance, but more advanced conditioning schemes have an advantage over simply feeding the loss parameters as one of the network inputs.

## B.3   STYLE TRANSFER

Figure 12 shows additional qualitative results on style transfer for different content and style images.

## C   THEORY

We start by proving Proposition 1 from the main paper and then comment on the cases when the conditions of the proposition are satisfied.

*Proof of Proposition 1.*  Denote the values minimized in optimization problems (1) and (2) by $V_1(\cdot)$ and $V_2(\cdot)$ respectively. From the definition of $F^*$ it follows that $V_1(G(\cdot, \lambda)) \geqslant V_1(F^*(\cdot, \lambda))$ for any $\boldsymbol{\lambda} \in \Lambda$ and $G \in C(\mathbb{X} \times \Lambda)$. Therefore, $V_2(G) \geqslant V_2(F^*)$, so $F^*$ minimizes (2). Thus, for any $G^*$ that minimizes (2):

$$0 = V_2(G^*) - V_2(F^*) = \mathbb{E}_{\boldsymbol{\lambda} \sim P_{\boldsymbol{\lambda}}} \left[ V_1(G^*(\cdot, \lambda)) - V_1(F^*(\cdot, \lambda)) \right]. \qquad (6)$$

Moreover, $V_1(G^*(\cdot, \lambda)) - V_1(F^*(\cdot, \lambda)) \geqslant 0$ for all $\boldsymbol{\lambda} \in \Lambda$. Expectation of a non-negative function is 0 only if it equals 0 almost surely, which means that $V_1(G^*(\cdot, \lambda)) = V_1(F^*(\cdot, \lambda))$ a.s., q.e.d.   □

The most restrictive condition of the proposition is the continuity of the function $F^*(\cdot, \cdot)$ as a function of $\lambda$. Indeed, generally $\arg\max$ is a multi-valued mapping that can be shown to be hemi-continuious using Berge's Maximum theorem (Berge, 1963) under weak technical conditions. However, there does not necessarily exist a continuous selection of this multi-valued mapping. Its existence can be proven under very restrictive conditions, such as the uniqueness of the minimizer,

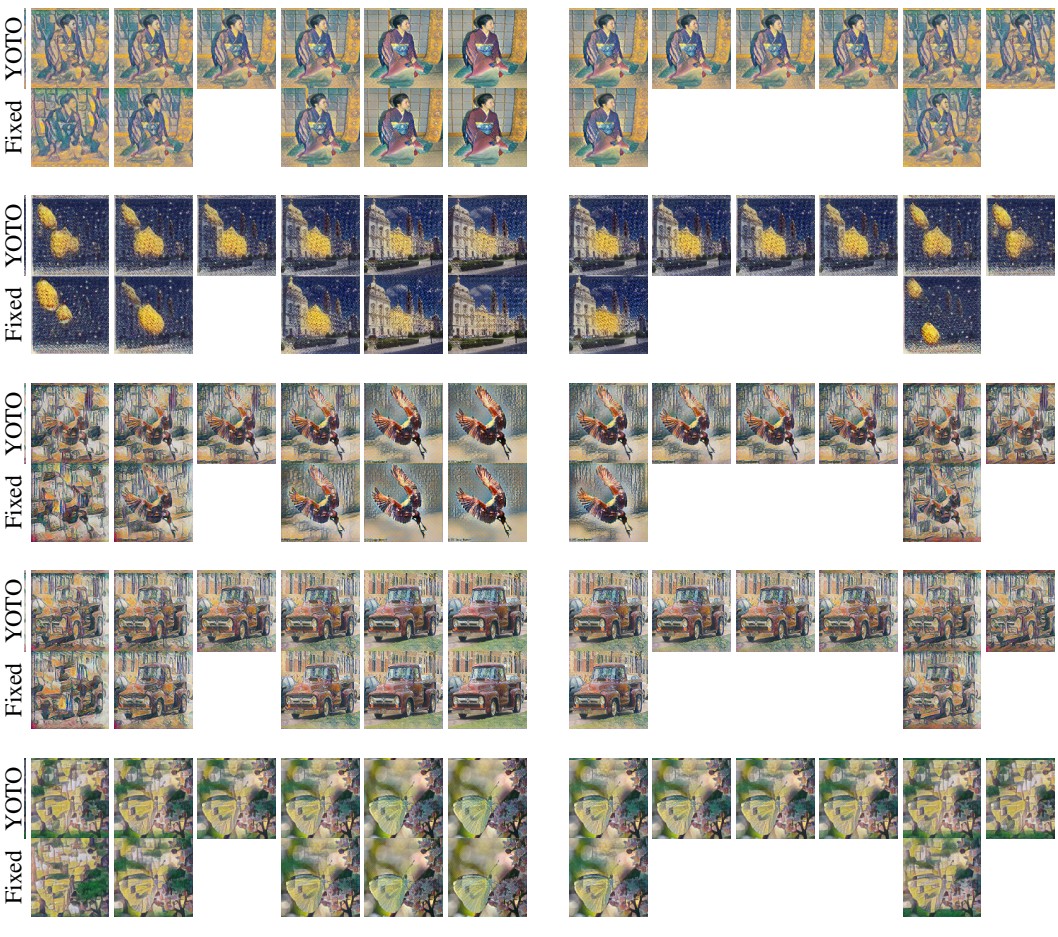

(a) Varying the content coefficient $\lambda_c$       (b) Varying one of the style coefficients $\lambda_{s,3}$

Figure 12: Additional qualitative comparison of image stylization models on images from the validation set of ImageNet. The first row shows the results of YOTO trained on all parameters, the second row – of models trained independently per configuration. In both cases parameter value increases from left ot right. YOTO results are very similar to those of independently trained models.

which clearly do not hold for optimization problems encountered in deep learning. While empirically the existence of a continuous $F^*$ seems natural, it remains to be seen if theoretical guarantees can be provided for realistically complex scenarios.

Another direction for extending Proposition 1 is explicitly incorporating function approximation. With function approximation, it would likely not be possible to prove that minimizers of the two problems coincide almost surely, but rather that they can be made arbitrarily close with a probability arbitrarily close to 1. Overall, we believe our work raises a range of interesting theoretical questions to be addressed in future research.

