# OpenReview forum: "You Only Train Once: Loss-Conditional Training of Deep Networks"
_ICLR.cc/2020/Conference — Accept (Poster)_

### Official Review · AnonReviewer5 · 2019-10-22
**Official Blind Review #5**

**Rating:** 6

**Review:**

Due to the late rebuttal, I was not able to respond during discussion time.

Q1, Q2) sound and look convincing.
Q3) I still cannot find details on this in the paper. How the validation set was chosen? What was the size? The authors need to make all the experiments fully reproducible.
Q4, Q5) ok

My concerns were sufficiently addressed in the current revision, and I will increase my score. However, the paper still feels close to a borderline, but probably, tending to "accept".

Also, I agree with Review #4, that also wondering about the application of the proposed method to hyperparameter search (Suggestion 1 in my review). Even if this would not be a sota in hyperparameter search, it feels like missing the opportunity to make the paper much stronger, by adding one more nice property to the proposed model.

-----

Generative models often use a loss function that is a weighted sum of different terms, e.g., data-term and regularizer. Let's denote these weights as λ. The paper proposes a method for learning a single model that approximates the result produced by a generative model for a range of loss-term weights λ. The method uses the following mechanisms i) λ-conditioned layers ii) training with a stochastic loss function, that is induced by a (log-uniform) distribution over λ. The performance of the model is demonstrated on the following problems learning β-VAE, image compression, and style transfer. The models clearly demonstrate an ability to approximate problem solutions for a range of coefficients. The paper is clearly written. The experiments, however, need future discussion.

1) The beta-VAE experiments (sec. 4.1)

Q1. While models demonstrate a reasonable behavior on Shapes3d dataset. The samples and reconstructions on CIFAR10 (Figure 8) indicate that all models are not trained well. If this is the case, conclusions might be misleading, since the approximating output of undertrained models might be much simple comparing to well-trained ones. Authors may want to provide a comparison of the trained models with conventional VAEs (with β=1), the reference figures for CIFAR10 are provided, for example, in https://arxiv.org/abs/1606.04934.

Q2. Wider YOTO seems to help a lot, but, what happens to the baseline models of increased size?

"We select the fixed β so that it minimizes the average loss over all β values."

Q3. Was it done directly on a test set, or were validation-data used?

2) Image compression (sec. 4.3)

"Finally, a wider model trained with a larger batch size (“YOTO wider batch16”) closely follows the fixed weight models in the high compression regime and outperforms them in the high quality regime." (Figure 5)

Q4. How is this compared to the baseline with batch16?

Q5. Authors also may want to provide std for provided metrics. The difference does not look statistically significant.

Suggestion 1: It might also be interesting to see if we can use this technique to perform a hyperparameter search. Train the model, select one the best performing set of hyperparameters, and then train models with this best value.

Overall, the paper proposes an interesting technique, that surprisingly, can work for a range of hyperparameters, and potentially have a high practical impact. However, the empirical evaluation is half-baked, specifically has certain methodological drawbacks e.g., perhaps undertrained beta-VAE model, absence of standard deviations while comparing (close) numerical results, and comparing models with different optimization parameters -- the performance difference might be due to optimization.

I recommend to reject the paper, however, I will appreciate discussions with authors and other reviewers, and will consider changing my score in case of reasonable argumentation.

**Experience Assessment:**

I have published one or two papers in this area.

**Review Assessment: Checking Correctness Of Derivations And Theory:**

I assessed the sensibility of the derivations and theory.

**Review Assessment: Checking Correctness Of Experiments:**

I assessed the sensibility of the experiments.

**Review Assessment: Thoroughness In Paper Reading:**

I read the paper at least twice and used my best judgement in assessing the paper.

---

> ### Author Response · Authors · 2019-11-15
> **Official response**
>
> We thank the reviewer for useful comments. We agree that the experimental evaluation had certain shortcomings, and we think the reviewer’s feedback allowed us to substantially improve it. We have uploaded an updated version of the paper addressing the issues raised. Further details are provided below.
>
> Q1. Overall, to our knowledge, vanilla VAEs are known to not perform very well on CIFAR-10. The model the reviewer pointed at, “Improved Variational Inference with Inverse Autoregressive Flow”, involves a few modifications, most notably a ResNet architecture with hierarchical latents, an associated inference scheme, as well as an inverse autoregressive flow posterior. It would be interesting to apply our method with this model, but in this paper for the sake of simplicity we experiment with a usual encoder-decoder convnet with one layer of latents. We verified that the networks are well converged. However, we have found that (perhaps unsurprisingly)  a substantial improvement in performance can be gained by simply increasing the dimensionality of the latent representation and increasing the network capacity: both the reconstructions and the samples become much sharper. We report the results with these higher-capacity architectures in the updated manuscript.
>
> Q2. Thanks for this question, we now performed a controlled experiment on the impact of network capacity on the performance of the method. To this end, we trained a set of networks with gradually increasing widths. The results are shown in Figure 2 and Figure 4. For small-capacity models, fixed-weight networks perform substantially better than YOTO. This is to be expected, since with limited capacity it is difficult for a single YOTO network to cover different parameter settings. However, when increasing the network capacity, YOTO catches up and almost matches the performance of per-weight trained networks.
>
> Q3. We performed this selection on the validation set.
>
> Q4. We agree that the batch16 models were confusing, and we removed them altogether.  Now both the fixed-weight and the YOTO models are trained in exactly the same way.
>
> Q5. We re-trained every model 3 or 4 times, and report standard deviations in most experiments in the updated paper. We found training to be quite stable on most tasks.

---

### Official Review · AnonReviewer4 · 2019-10-24
**Official Blind Review #4**

**Rating:** 6

**Review:**

The problem tackled by the paper is related to the sensitivity of deep learning models to hyperparameters. While most of the hyperparameters correspond to the choice of architecture and optimization scheme, some influence the loss function. This paper assumes that the loss function consists of multiple weighted terms and proposes the method of finding the optimal neural network for each set of parameters by only training it once.

The proposed method consists of two aspects: the conditioning of the neural network and the sampling of the loss functions' weights. Feature-wise Linear Modulation is used for conditioning and log-uniform distribution -- for sampling.

My decision is a weak accept.

It is not clear to me if the choice of performance metrics is correct. In many practical scenarios, we would prefer a single network that performs best under a quality metric of choice (for example, perceptual image quality) to an ensemble of networks that all are good at minimizing their respective loss functions. Therefore, the main performance metric should be the following: how much computation is required to achieve the desired performance with respect to a chosen test metric.

Moreover, it might be obvious that the proposed method would be the best w.r.t. this metric, compared to other hyperparameters optimization methods, since it only requires a neural network to be trained once with little computational overhead on top. But then its performance falls short of the "fixed weight" scenario, where a neural network is trained on a fixed loss function and requires to raise the complexity of the network to achieve similar performance.

Therefore, obtaining a neural network that would match the desired performance in the test time and would have a similar computational complexity requires more than "only training once", with more components, such as distillation, required to be built on top of the proposed method. The title of the paper is, therefore, slightly misleading, considering its contents.

Also, it is slightly disappointing that the practical implementation of the method does not allow a more fine-grained sampling of weights, with uniform weights sampling shown to be degrading the performance. This implies that the method would have to be either applied multiple times, each time searching for a more fine-grained approximation for the best hyperparameters, or achieve a suboptimal solution.

Below are other minor points to improve that did not affect the decision:
-- no ImageNet experiments for VAE
-- make plots more readable (maybe by using log-scale)
-- some images are missing from fig. 7 comparison

**Experience Assessment:**

I have read many papers in this area.

**Review Assessment: Checking Correctness Of Derivations And Theory:**

I did not assess the derivations or theory.

**Review Assessment: Checking Correctness Of Experiments:**

I assessed the sensibility of the experiments.

**Review Assessment: Thoroughness In Paper Reading:**

I read the paper at least twice and used my best judgement in assessing the paper.

---

> ### Author Response · Authors · 2019-11-15
> **Official response**
>
> We thank the reviewer for the positive feedback and useful suggestions. Below we comment on the raised concerns.
>
> We respectfully disagree that the focus should always be on maximizing a single metric. While in some applications this may be desirable, in others it is important to obtain a family of models that covers the full loss frontier. For instance, this is the case both for image compression and style transfer: it is desirable to be able to vary the parameters — the compression rate and the degree of stylization, respectively — at inference time. We do agree that for some other tasks only a single best-performing model may be of interest, in which case the proposed metric would be useful.
>
> We agree that it would be interesting to fine-tune a YOTO-trained model with a fixed-weight loss to get a further improvement in performance. We were not able to perform this experiment during the rebuttal time, but will strive to include them in the final paper. However, in the cases we study, even the models trained without such fine-tuning can perform very close to the single-weight models, especially if the YOTO model has higher capacity than the fixed-weight models.
>
> We are unfortunately unsure about how to exactly interpret the comment about weight sampling. Indeed we have found that sampling from a uniform distribution over the weights does not perform very well, but log-uniform worked well in our experiments, even if the sampling range was quite large, up to 3 orders of magnitude. Re-training with a narrower range could indeed further improve the results, but we are unsure what this has to do with uniformity vs log-unformity of the weight distribution.
>
> Comments about the minor points:
> - It would be interesting to also report VAE results on ImageNet, but VAEs are not commonly evaluated on ImageNet, so we choose to stick to more standard datasets and rather perform more in-depth experiments on these.
> - We have done our best making the plots as readable as possible, including plotting both the frontiers and the full losses, subtractive normalization of the full losses, and experimenting with log-scales of the axes (the latter unfortunately did not always improve the readability of the plots). The plots currently presented in the paper are our best attempt. We would be grateful for specific advice on improving the plots.
> - Some images are missing in Fig. 7 because producing each of the images in the bottom row requires training an additional fixed-weight model, and we chose to only train a subset of them to save computation.

---

### Official Review · AnonReviewer2 · 2019-11-04
**Official Blind Review #2**

**Rating:** 6

**Review:**

Summary
-------------
The authors propose a methodology to train a single Deep Neural Network (DNN) that minimize a parametrized (weighted) sum of different losses. Since the model is itself conditioned by the weights, it allows to train a single model for all weights values, instead of retraining when the weights change.

Experiments suggest that this methodology does not degrade the performances to much w.r.t. retraining on every weight changes.

The proposed conditioning of the layers is done via a reparametrization (FiLM, Perez et al. 2018) of the weights with a scale $\sigma(\lambda)$ and a bias $\mu(\lambda)$ where $\mu$ and $\sigma$ are MLP. This allows to condition the layer on $\lambda$, while keeping the number of parameters low.

Novelty
----------
The idea of integrating a family of loss functions with model conditioning has also been proposed by Brault et al. [1], in the context of multi-task kernel learning. Hence I believe this work should be acknowledged.

As the product of kernels is the tensor product of their feature maps, it would suggest to condition the network's layers by taking the tensor product of the weights with respect to an MLP on $\lambda$. This could be applied on each layers or simply on the last Fully Connected layer. Note however that it would drastically increase the number of parameters and hence not be a viable solution (or maybe with some channel pooling?).

References:

[1] Infinite Task Learning in RKHSs; Brault, Romain and Lambert, Alex and Szabo, Zoltan and Sangnier, Maxime and d'Alche-Buc, Florence; Proceedings of Machine Learning Research; 2019.

Quality
----------
The paper is self content and well written.

Experiments are well detailed and seems to be reproducible. I would be a great addition to release the code in a public repository (with a link in the paper or appendices) if the paper is accepted.

I would also suggest the following experiments:
    * An experiment showing the time penalty induce by training the loss conditional model. The authors claims that training multiple separate models is inefficient compared to their proposed method. While it seems obvious, it deserve an experiment as one of the claim.
    * The authors propose to sample one $\lambda$ per SGD iteration. However it ma be useful to sample more of them. Especially when the set of $\lambda$ is large (high dimensional)
    * Possibly use a pre-trained model and only tune the $\sigma(\lambda)$, $\mu(\lambda)$ MLPs

Overall my decision is weak accept, the paper lacks of novelty and the experiments could be more extensive.

**Experience Assessment:**

I have published one or two papers in this area.

**Review Assessment: Checking Correctness Of Derivations And Theory:**

I carefully checked the derivations and theory.

**Review Assessment: Checking Correctness Of Experiments:**

I assessed the sensibility of the experiments.

**Review Assessment: Thoroughness In Paper Reading:**

I read the paper at least twice and used my best judgement in assessing the paper.

---

> ### Author Response · Authors · 2019-11-15
> **Official response**
>
> We thank the reviewer for the positive feedback  and useful suggestions.
>
> We acknowledged the work of Brault et al. in the updated version of the manuscript.
>
> We will release code associated with the paper after it is accepted for publication.
>
> The proposed experiments are indeed interesting. We now comment on each one separately:
> 1) Timing results are provided in the appendix of the updated manuscript. When using the same network architecture, the proposed method slows down training by only 8%.
> 2) There might have been some misunderstanding here. In most of our experiment we do sample the weights per training sample, not per SGD iteration. We have also tried sampling weights per mini-batch and observed the performance is very close, but on average slightly worse. We added this model to the ablation study in the appendix.
> 3) We agree that his would be a very interesting experiment to run, but unfortunately we were not able to complete it during the rebuttal period. We will strive to do it for the final version of the paper.

---

### Public Comment · ~Vincent_Dumoulin1 · 2019-10-17
**Relevant work**

The submission is a really interesting application of feature-wise transformations. There exists prior work on using conditional instance normalization to condition a style transfer network on content and style loss coefficients (Babaeizadeh and Ghiasi, 2019) that should be acknowledged. As a result, I don’t think this submission can claim novelty on the idea of conditioning a network on loss coefficients, but it is still a valuable contribution in that it demonstrates the general applicability of this idea beyond the style transfer domain.

References:

Babaeizadeh, M. & Ghiasi, G. (2019). Adjustable Real-time Style Transfer. In ICLR Workshop on Deep Generative Models for Highly Structured Data.

---

> ### Author Response · Authors · 2019-10-23
> **Thanks**
>
> Thanks for pointing out this very relevant paper, unfortunately we missed it during the search for related work. We will properly cite it and otherwise modify our submission accordingly.

---

### Author Response · Authors · 2019-11-15
**Paper revision**

We have uploaded an updated version of the paper. Here are the main changes:
- Cited Brault et al. and Babaeizadeh & Ghiasi
- Re-ran all models at least 3 times and reported standard deviations
- Ran beta-VAE experiments with models of varying capacity, by changing the width of the networks. The results are reported in Figures 2 and 4
- Increased the latent state size to 256 for VAEs on CIFAR-10, resulting in better qualitative and quantitative results
- Timed the proposed method compared to a fixed-weight model. The proposed method is only 8% slower
- Removed batch16 models from the compression experiment and trained all compression models for 2 million iterations
- Added a model with per-batch sampling of loss weights to the ablation study

---

### Decision · Program_Chairs · 2019-12-19

**Decision:**

Accept (Poster)

**Comment:**

The paper proposes and validates a simple idea of training a neural network for a parametric family of losses, using a popular AdaIN mechanism.
Following the rebuttal and the revision, all three reviewers recommend acceptance (though weakly). There is a valid concern about the overlap with an ICLR19-workshop paper with essentially the same idea, however the submission is broader in scope and validates the idea on several applications.